# $\mathcal{E}$XT$\mathcal{P}$OSE
# Robust and Coherent Pose Estimation by Extending ViTs

**Rongyu Chen** [*†1]  **Li'an Zhuo** [*2]  **Linlin Yang** [3]  **Qi Wang** [2]  **Liefeng Bo** [2]  **Bang Zhang** [2]  **Angela Yao** [1]

https://gloryyrolg.github.io/extpose

## Abstract

Vision Transformers (ViT) are remarkable at 3D pose estimation, yet they still encounter certain challenges. One issue is that the popular ViT architecture for pose estimation is limited to images and lacks temporal information. Another challenge is that the prediction often fails to maintain pixel alignment with the original images. To address these issues, we propose a systematic framework for 3D pose estimation, called $\mathcal{E}$XT$\mathcal{P}$OSE. ExtPose extends image ViT to the challenging scenario and video setting by taking in additional 2D pose evidence and capturing temporal information in a full attention-based manner. We use 2D human skeleton images to integrate structured 2D pose information. By sharing parameters and attending across modalities and frames, we enhance the consistency between 3D poses and 2D videos without introducing additional parameters. We achieve state-of-the-art (SOTA) performance on multiple human and hand pose estimation benchmarks with substantial improvements to 34.0$mm$ (**-23%**) on 3DPW and 4.9$mm$ (**-18%**) on FreiHAND in PA-MPJPE over the other ViT-based methods respectively.

> "You can enjoy a grander sight,
> By climbing to a greater height."

*Tang Poems*

## 1. Introduction

Human and hand pose estimation is a foundation for higher-level applications in robotics, action recognition, animation,

[†]Work done during internship at Alibaba Group [*]Equal contribution [1]Computer Vision & Machine Learning Group, National University of Singapore [2]Tongyi Lab, Alibaba Group [3]Communication University of China. Correspondence to: Bang Zhang <zhangbang.zb@alibaba-inc.com>.

*Proceedings of the $42^{nd}$ International Conference on Machine Learning*, Vancouver, Canada. PMLR 267, 2025. Copyright 2025 by the author(s).

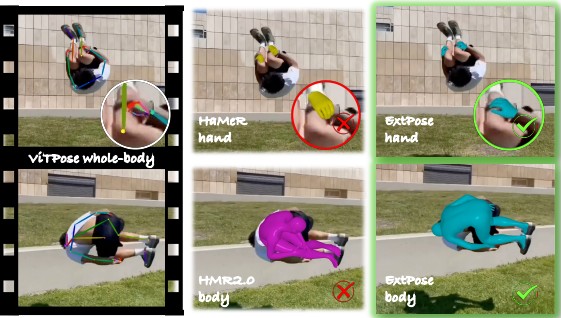

Figure 1: **ExtPose extends and aids SOTA ViT-based works in generalization with auxiliary 2D detection ($1^{st}$ column).** Our method ($3^{rd}$ column) excels in robust and pixel-aligned mesh reconstruction even in extremely complicated parkour while ViT-based works ($2^{nd}$ column) predict a completely flipped hand and body.

human-object interaction, AI generation, *etc*. With the extensive use of vision transformers (ViT) (Dosovitskiy et al., 2021), there has been a growing trend towards leveraging ViT architectures for 2D HPE[1] (Xu et al., 2022) and 3D HPE (Goel et al., 2023; Pavlakos et al., 2024). These methods boast remarkable results after training on large-scale datasets, but still face some challenges with robust generalization and temporal coherence. Specifically, they struggle to align with 2D image evidence; orthogonally, when encountering data in more common video formats, they cannot incorporate temporal information, instead merely guessing jittering poses frame by frame under depth and occlusion ambiguity.

*First*, predictions are often inconsistent with input images, especially under challenging settings such as occlusions (Chen et al., 2023), motion blur (Oh et al., 2023; Chen et al., 2025b), poor lighting, and other imaging degradations (Dwivedi et al., 2024a). For instance, in the middle column of Fig. 1, the body estimate is oriented wrongly in the camera. In the case of hand pose estimation, the issue becomes more severe, as the hand tends to occupy only a

---
[1]For simplicity, HPE and PE in this work are used interchangeably and denote Human and Hand Pose Estimation unless otherwise specified.

small region in the capture and is flexible to move in space. Additionally, the palm and back of the hand have similar textures, leading to incorrect wrist rotations that globally flip the hand (Fig. 1). The challenging alignment issue of 3D pose estimation is not resolved by simply upgrading to the ViT architecture but also related to the task itself. Our method improvement is inspired by the observation that 2D HPE models show better 2D alignment than those trained on 3D HPE tasks with the same architecture.

*Second*, the majority of ViT-based works focus on single-frame inference. With video inputs, however, temporal information is highly beneficial for handling complex occlusions and improving prediction stability. Adding the temporal dimension into ViT architectures incurs substantial computational overhead and demands 3D labeled video for training, thus impeding the development of video ViT-based methods in this domain. As a result, existing video-based methods (Shen et al., 2023; Shin et al., 2024) rely on static frame features and additional temporal modules, thereby not fully maximizing the spatiotemporal feature interactions at each layer. Our approach is to extend the well-developed image-based ViT to the video version along with advancements in large model training (OpenAI, 2024; Rasley et al., 2020).

This work enhances the image alignment and temporal coherence of the ViT-based HPE features through modular attention extension and a unified framework. We draw inspiration from the use of 2D skeletons as control signals in conditional human image and video generation (Zhang et al., 2023a; Hu et al., 2024). Moreover, the use specifically of 2D skeleton *images*, which precisely depict the human kinematic chain. Furthermore, the 2D skeleton images exhibit strong pixel-wise spatial consistency with the original image. This enables joint processing with the input image with a shared backbone and can facilitate the use of pre-trained models. Extending this idea to the temporal dimension, each frame of a video can also be processed by a shared ViT backbone in parallel, with temporal relationships captured by frame interactions at each layer.

To that end, we propose EXTPOSE, short for **Ext**ending **Pose**s, which *extends* a pre-trained ViT backbone with an *extended* attention mechanism. EXTPOSE elegantly leverages the power of ViT attention to model arbitrary and long-range relationships and flexible pairwise computations, inherently suitable for video tasks. It also allows for extensions to attend across different image and 2D pose modalities. Specifically, the attention mechanism is applied in a unified and hierarchical manner, depending on the availability of input information, and integrates auxiliary 2D pose evidence and video context to promote features for HPE. Finally, the framework does not introduce extra parameters or modules, as existing works do (Shin et al., 2024), and

reuses pre-trained weights as much as possible.

Our unified EXTPOSE achieves remarkable improvement over previous ViT-based methods. For instance, with the aid of 2D hand pose estimates, the model has a better sense of hand global orientation (Fig. 1). Using video frames as input with temporal information alleviates prediction flickers and consistently gains in PA-MPJPE. With the thought of unifying representation and reusing pre-trained knowledge, the model is enhanced at a rapid convergence. Additionally, we evaluate thoroughly the effectiveness of key design components, including 2D pose representations, fusion strategies, and learning capabilities. The framework is effective on the human body and hands for both image- and video-based settings. It binds these relevant settings with the foundational ViT-based methods; thus, they can benefit from advances made in ViT-based methods.

We summarize our contributions as follows:

- we propose the innovative and systematic EXTPOSE framework to extend and enhance attention and ViT-based HPE backbones. The effectiveness is demonstrated by addressing two concrete issues in image alignment and temporal coherence in this work.

- for effective fusion and utilization of the auxiliary 2D pose, its representation is unified and interacts with the image via the proposed dual-stream cross-modal attention of a shared ViT.

- a new method is introduced for video-based HPE, where temporal context is seamlessly integrated through an extension of 2D image attention to 3D video attention, obviating the need for additional temporal modules.

- EXTPOSE effectively remedies the current SOTA ViT-base method and promisingly shows consistent state-of-the-art results on several hand and human benchmarks including both monocular image and video settings.

## 2. Related Work

### 2.1. Monocular Human & Hand Pose Estimation

**Image-based HPE.** 3D pose and shape estimation methods commonly use parametric 3D mesh models such as SMPL (Loper et al., 2015). SMPL-based HPE methods (Kanazawa et al., 2018; Kolotouros et al., 2019; Lin et al., 2021; Li et al., 2022; Wang et al., 2023) use a CNN backbone (*e.g.*, ResNet, HRNet) to extract image features and then regress pose and shape coefficients. More recent human (Goel et al., 2023; Dwivedi et al., 2024b; Zhuo et al., 2023) and hand (Pavlakos et al., 2024; Potamias et al., 2025) pose estimation methods have adopted a ViT architecture (Dosovitskiy et al., 2021). MultiHPE (Baradel et al., 2024) and

AiOS (Sun et al., 2024) build on ViT for multi-task learning in a bottom-up paradigm, including human detection and pose estimation. Given ViT's potential for simplicity, scalability, and superior performance in HPE, it provides a strong foundation for our work.

**Video-based HPE** extracts static features with CNN-based models (Kolotouros et al., 2019) and then use CNNs (Kanazawa et al., 2019), RNNs (Kocabas et al., 2020; Choi et al., 2021; Tekin et al., 2019), or transformersWan et al. (2021); Shen et al. (2023); Fu et al. (2023) to model temporal relationships. For example, TCMR (Choi et al., 2021) segments sequences into past, future, and whole segments to ensure motion consistency. Wei et al. (2022) incorporate non-local attention and hierarchical feature fusion, while Wan et al. (2021); Shen et al. (2023); Fu et al. (2023) move to standard transformers (Vaswani et al., 2017). Despite temporal integration, recent works find video-based methods often "surprisingly" underperform image-based methods due to an over-smoothing problem (Shin et al., 2024; Shen et al., 2023). Recently, Shin et al. (2024) integrates motion cues like 2D ViTPose (Xu et al., 2022) and SLAM camera data to improve video-based HPE. In contrast, our work enhances image-based ViT HPE with 3D attention. It better interacts and fuses spatial-temporal information in a unified space and scalable manner (OpenAI, 2023; 2024), thereby eliminating the need for separate temporal modules and prerequisites, and achieving improved temporal coherence.

### 2.2. HPE with Auxiliary 2D Information

Different from fitting and adaptation with additional 2D (Kolotouros et al., 2019; Lin et al., 2025), efforts to improve pixel alignment in HPE regression during generalization have focused on incorporating 2D inference alongside the image. Boukhayma et al. (2019) concatenate a hand keypoint heatmap with the RGB image along the channel. More recently, Shin et al. (2024) follow the lifting paradigm (App. B) to integrate motion context from 2D joint coordinates via an MLP. Another approach learns spatially aligned intermediate representations. For example, Iqbal et al. (2018) use a 2.5D representation, combining 2D heatmaps and depth maps to predict 3D coordinates. Zhang et al. (2021); Kocabas et al. (2021); Potamias et al. (2025) employ multi-task learning, supervising 2D auxiliary tasks (*e.g.*, segmentation) to guide intermediate representations. These aligned representations often serve as spatial attention weights (Kocabas et al., 2021) or location features (Zhang et al., 2021; Potamias et al., 2025) during aggregation. As intermediate constraints tighten, they are incorporated into a cascading two-stage paradigm (2D task followed by lifting) (Sengupta et al., 2021), where various 2D representations act as proxies (Pavlakos et al., 2018). This work incorporates auxiliary 2D information as model input, thoroughly integrating it with the image through structured representation and extended attention.

## 3. Preliminaries

### 3.1. Hand & Human Parametric Models

The parametric human body model SMPL (Loper et al., 2015) and hand model MANO (Romero et al., 2017) is parameterized by the pose $\theta \in \mathbb{R}^{|\theta| \times 3}$ ($|\theta| = 72, 48$ for human and hand respectively) and shape $\beta \in \mathbb{R}^{10}$. It uses Linear Blend Skinning function $\mathcal{M}$ to estimate the 3D mesh with vertices $V \in \mathbb{R}^{|V| \times 3}$:

$$V, J = \mathcal{M}(\theta, \beta). \tag{1}$$

The pose $\theta$ consists of the global rotation of the root joint and the $|J| - 1$ local rotations of other joints relative to their parents along the kinematic tree. The 3D joints $J^{\text{3D}}$ are based on a linear combination of the vertices $V$; with camera projection parameters $c \in \mathbb{R}^3$, the 3D joints can be projected onto the 2D plane with projection $\Pi$:

$$J^{\text{2D}} = \Pi(J^{\text{3D}}, c). \tag{2}$$

### 3.2. Self-Attention & Vision Transformer

Image feature extractors have evolved from ResNet (He et al., 2016), HRNet (Sun et al., 2019) to ViT (Dosovitskiy et al., 2021). The ViT is scalable and stacks several of the same blocks to process information. It treats images as a series of $M$ non-overlapping $d \times d$ patch units $X_i \in \mathbb{R}^{d \times d \times 3}, i = 1, 2, \ldots, M, M = \frac{H}{d} \times \frac{W}{d}$ ($d = 16$ as default). The patches are first linearly projected into embeddings by $F_i = \text{PatchEmb}(X_i) \in \mathbb{R}^D$, where $D$ is the embedding dimension. To encode the patch's position in the original image, $\frac{H}{d} \times \frac{W}{d}$ learnable positional embeddings $\text{PosEmb}(i) \in \mathbb{R}^D$, with equal dimensions, are then added to the tokens $F_i$, *i.e.*, $F_i = F_i + \text{PosEmb}(i)$. In each subsequent block of the encoder, each token fully exchanges information with others with a core **Self-Attention (SA)** mechanism[2]:

$$F_i = F_i + \text{Attn}(F_i, \{F_i\}_M). \tag{3}$$

Formally, token $F_i$ computes attention weights $A_i$ with all tokens including itself by first obtaining the query, key, and value with respective projections $Q_i = W^Q F_i, K_i = W^K F_i, V_i = W^V F_i \in \mathbb{R}^D$. The weight $A_i \in \mathbb{R}^M$ is calculated by a normalized scaled dot-product operation based on pairwise similarity:

$$A_i = \text{Softmax}\left(\frac{[K_1, K_2, \ldots, K_M]^T Q_i}{\sqrt{D}}\right). \tag{4}$$

---

[2]The index $i$ in $\{F_i\}_{i=1}^{M}$ is omitted for simplicity.

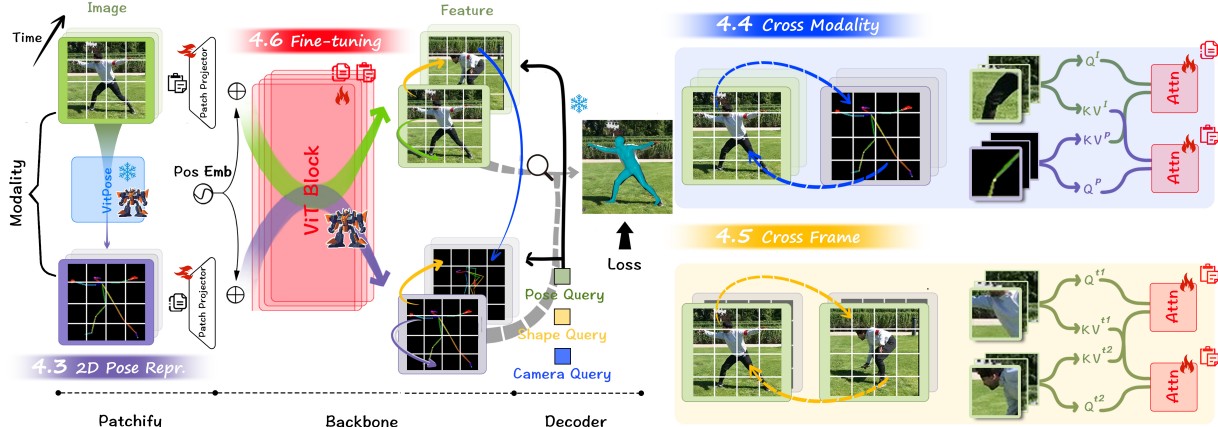

Figure 2: **The overview of EXTPOSE, an extension of SOTA self-attention and ViT-based backbone.** The skeleton image is used to represent 2D poses estimated by off-the-shelf ViTPose (Sec. 4.3). All ViT blocks are trainable and parameters are shared across the sequence of the image and 2D pose modality (Sec. 4.6). The image and 2D pose interact thoroughly with each other through cross-modal attention (Sec. 4.4). Besides, cross-frame attention facilitates different frames to communicate and propagate video temporal context (Sec. 4.5). The enhanced feature is decoded into the mesh supervised by the HPE losses in Eqs. (12) to (15).

The feature residual for token $i$ is then derived from aggregating values $\{V_i\}_M$ of all tokens with the attention weight $A_i$, *i.e.*, $A_i \odot [V_1, V_2, \ldots, V_M]$.

The above computation shows a core advantage of attention. Each token can collect information from tokens in any other position across the image; this facilitates capturing relationships between keypoints which are otherwise far away, *e.g.* between two wrists. Additionally, a FeedForward Net (FFN) usually follows to transform feature space at the end of the block. After stacked blocks, the feature of all patch tokens $\{F_i\}_M$ is extracted and could be applied to downstream tasks.

## 4. EXTPOSE

### 4.1. Problem Formulation

Consider a sequence of $T$ images of size $H \times W$ $\{I^t\}_T \in \mathbb{R}^{T \times H \times W \times 3}$. Additionally, consider a sequence of corresponding 2D poses $\{P^t\}_T$, and it becomes an image-based task when $T = 1$. The dimension of 2D poses $P^t$ which is annotated by an off-the-shelf 2D keypoint detector or human, varies from representations discussed in Sec. 4.3. We aim to learn an HPE model that predicts the pose parameters $\{\hat{\theta}^t\}_T \in \mathbb{R}^{T \times |\hat{\theta}|}$ for each frame and a video-shared shape parameter $\hat{\beta} \in \mathbb{R}^{10}$. The 3D mesh $\{\hat{V}^t\}_T \in \mathbb{R}^{T \times |\hat{V}| \times 3}$ and joint coordinate $\{\hat{J}^t\}_T \in \mathbb{R}^{T \times |\hat{J}| \times 3}$ could be obtained by Eq. (1). Camera projection parameters $\{c^t\}_T \in \mathbb{R}^{T \times 3}$ are also predicted to re-project the 3D mesh back onto the 2D plane (Eq. (2)), expected to align with human and hand in the image.

### 4.2. Framework Overview

EXTPOSE refines ViT-based HPE by improving image alignment and temporal coherence through a streamlined yet effective design. It first represents images, 2D poses, and video as unified visual token sequences $\{\{F_i^t\}_M\}_T$ within ViT (Sec. 4.3). This allows ViT to extract 2D pose features that encapsulate both spatial location and skeleton structure. To enhance localization, we introduce a cross-modal extension of self-attention, enabling thorough image-2D integration within a dual-stream design (Sec. 4.4). Orthogonally, Sec. 4.5 extends ViT's temporal scope by arranging tokens to attend across frames. Leveraging a pre-trained 3D ViT pose estimator, EXTPOSE ensures efficient learning with rapid convergence (Sec. 4.6).

### 4.3. Unified 2D Pose Feature Extraction

Various representations exist for 2D poses beyond traditional joint coordinates used in the HPE work (Zhang et al., 2022), including **heatmaps** (Sun et al., 2019) and **rendered skeleton images** (Zhang et al., 2023a) (Fig. 2). Heatmap representations $\boldsymbol{H} \in \mathbb{R}^{H \times W \times |J|}$ place Gaussians at joint locations spatially but are inconsistent in channels within diverse datasets and lack explicit structural relationships. While the skeleton images $I^p \in \mathbb{R}^{H \times W \times 3}$ depict stick-figure poses on a blank background, naturally aligning with human perception like RGB images. Joints are rendered as distinctively colored circles, connected by gradient-colored bones reflecting kinematic structure, with confidence (Gu et al., 2024) levels encoded via alpha blending. Confidence helps threshold unreliable 2D pose detection.

To integrate 2D poses into ViT's visual domain, inspired by (Zhang et al., 2023b), we employ skeleton images as an alternative 2D pose representation. These images undergo the same patch embedding, positional encoding, and SA encoder processing as described in Sec. 3.2.

Training with skeleton images achieves rapid convergence and strong results without modifying architecture or training settings (Sec. 5.4). This underscores ViT's generalizability and scalability while demonstrating that 2D pose features extracted this way align well with image features. By employing a unified token-based framework, EXTPOSE maintains flexibility for future HPE model advancements.

### 4.4. Dual-stream Image-2D Pose Fusion

Figure 2 illustrates the dual-stream EXTPOSE-dual architecture, where a shared ViT backbone processes parallel feature streams: image features $\{F_i^I\}_M$ and 2D pose features $\{F_i^p\}_M$. Both Image and 2D pose modality attend to both intra- and cross-modal features, ensuring comprehensive fusion. For an image feature $F_i^I$, the attention weights for self and cross-modal information collection are computed as:

$$A_i^I = \text{Softmax}\left(\frac{[S_i^{I\text{-}I}; S_i^{I\text{-}p}]}{\sqrt{D}}\right), \tag{5}$$

$$\text{where } S_i^{I\text{-}I} = K^I Q_i^I, \quad S_i^{I\text{-}p} = K^p Q_i^I \in \mathbb{R}^M. \tag{6}$$

$K, Q, V$ are computed following Sec. 3.2.

Similarly, the 2D pose feature also attends to both image and pose streams:

$$A_i^p = \text{Softmax}\left(\frac{[K^I; K^p] Q_i^p}{\sqrt{D}}\right). \tag{7}$$

The final features enhanced by the other new branch information are:

$$\Delta F_i^I = A_i^I \odot V^{Ip}, \quad \Delta F_i^p = A_i^p \odot V^{Ip}, \tag{8}$$

where $V^{Ip} = [V^I; V^p]$. The whole operation is denoted as:

$$[F_i^I, F_i^p] = [F_i^I, F_i^p] + \text{Attn}([F_i^I, F_i^p], \{F_i^I, F_i^p\}_M), \tag{9}$$

and can be implemented by concatenating the two streams before self-attention.

Note that the bimodal SA in a shared space benefits from our unified image-2D pose feature representation. Dot product-based attention assembles information anywhere in space without requiring perfectly aligned 2D poses. It maximizes the interaction between image and 2D pose features, allowing the network to refine features of both modalities dynamically and adaptively. As EXTPOSE-lift justifies easy model adaptation to 2D pose, learning more robust predictions is expected to converge fast.

### 4.5. Attention from Images to Videos

Without requiring additional temporal modules, we extend EXTPOSE with its inherent attention mechanism to handle the temporal dimension. Unlike current video-based HPE methods, which use static frame features from image-based backbones (Kolotouros et al., 2019) and learn motion on top, we directly leverage unified ViT's patchified tokens, enabling access to richer information, *i.e.*, $F_i^t, i = 1, 2, \ldots, M, t = 1, 2, \ldots, T$, where $T$ denotes the number of frames. Furthermore, while CNNs and RNNs struggle to capture long-term spatiotemporal dependencies, self-attention (SA) excels by allowing computation across frames, not just within them. This enables batch processing of frames, allowing information to propagate across distant frames, thereby enhancing temporal coherence and reducing monocular ambiguity in features and predictions.

Specifically, beyond inter-frame attention (Eq. (3)) between $F_i^{t_1}$ and $\{F_i^{t_1}\}_M$, a token $F_i^{t_1}$ containing a joint could attend to the same spatial location $F_i^{t_2}$ in a different frame $t_2$ and new location $j$ at the different time $F_j^{t_2}, j \neq i$ to capture motion. Note that the same location at $t_2$ may cover the background.

The 3D spatiotemporal attention on inflated features $\{\{F_i^t\}_M\}_T$ is formulated as:

$$F_i^t = F_i^t + \text{Attn}(F_i^t, \{\{F_i^t\}_M\}_T). \tag{10}$$

Above, $F_i^t$ without superscript "$I$" and "$p$" implies the 3D attention is only performed *within respective modality streams*, *i.e.*, orthogonal to the cross-modal attention in Sec. 4.4, as it aims to focus on the time dimension, which can be implemented by simply collapsing the modality axis into the batch axis or versatile attention mask (Vaswani et al., 2017). At each attention block, for two stream feature $\{\{F_i^{tI}\}_M\}_T, \{\{F_i^{tp}\}_M\}_T \in \mathbb{R}^{T \times M \times D}$ concatenated along the modality axis, the attention mask matrix has the following structure:

$$M = \begin{bmatrix} \mathbf{0}_{T \times M} & -\infty_{T \times M} \\ -\infty_{T \times M} & \mathbf{0}_{T \times M} \end{bmatrix}, \tag{11}$$

where $M \in \mathbb{R}^{2TM \times 2TM}$ is added into correlation within the Softmax operation Eq. (4) so that $-\infty$ masks out inactive interaction across modalities after the Softmax operation.

This zero-shot "free lunch" for enhancing temporal coherence even comes with directly performing 3D attention on a pre-trained image-based model. Fusion in the face of disagreement benefits from further training to advance performance, with additional temporal positional embeddings that encode frame location and motion ordering.

## 4.6. Training Losses & Implementation Details

We follow Kolotouros et al. (2019); Goel et al. (2023) and train on both 3D and 2D data jointly. The total loss $\mathcal{L}$ is a weighted sum of the losses in Eqs. (12) to (15). These include a 3D joint loss $\mathcal{L}_{\text{joint}}$, parameter loss $\mathcal{L}_{\text{param}}$ when SMPL parameters are available, a 2D reprojection loss $\mathcal{L}_{\text{reproj}}$, and an adversarial loss with a discriminator net $D$ to impose pose and shape feasibility (Kanazawa et al., 2018):

$$\mathcal{L}_{\text{joint}} = \|\hat{J}^{3D} - J^{3D}\|_1, \tag{12}$$

$$\mathcal{L}_{\text{param}} = \|\hat{\theta} - \theta\|_2^2 + \|\hat{\beta} - \beta\|_2^2, \tag{13}$$

$$\mathcal{L}_{\text{reproj}} = \|\hat{J}^{2D} - J^{2D}\|_1, \tag{14}$$

$$\mathcal{L}_{\text{adv}} = \|D(\theta, \beta) - 1\|_2^2. \tag{15}$$

During the training of all backbone parameters, besides standard affine and color data augmentation (Goel et al., 2023), each modality is masked out as a whole with a probability of 50% to cultivate EXTPOSE's ability to extract features in each input modality individually. We use the PyTorch implementation of scaled dot product attention with the mask and accelerated flash attention to speed computation and save GPU memory. An AdamW optimizer (Loshchilov & Hutter, 2019) is deployed with a learning rate 1e-5, $\beta_1 = 0.9$, $\beta_2 = 0.999$, and a weight decay of 1e-3. Training lasts for 50K iterations with a batch size of 32 on 8 A100 GPUs. For other details of datasets and settings, please refer to the human and hand experiment Secs. 5.2 and 5.3.

## 5. Experiments

### 5.1. Evaluation Metrics

For *3D accuracy* evaluation, the reconstruction performance is usually measured in terms of Mean Per Joint and Vertex Error (**MPJPE**, **MPVPE**, in $mm$) and the ones after Procrustes Alignment (**PA-MPJPE**, **PA-MPVPE**). Additionally, the F-score of correct poses with errors less than $5mm$ and $15mm$ (**F@5**, **F@15**) is evaluated for hand benchmarks (Hampali et al., 2020). For *2D pose alignment* evaluation, the commonly used Percentage of Correct Keypoint (**PCK**) metric is computed at different error tolerance thresholds including 0.05, 0.1, and 0.15.

### 5.2. Human Pose Estimation

**Settings.** Following standard practice (Shin et al., 2024; Goel et al., 2023), EXTPOSE initialized from HMR2.0 is trained on mixed 3D datasets including 3DPW (Von Marcard et al., 2018), Human3.6M (Ionescu et al., 2013), MPI-INF-3DHP (Mehta et al., 2017), and COCO (Lin et al., 2014).

**Image benchmark.** We first evaluate our method in image-

Table 1: **Comparison with SOTA HPE methods on the 3DPW dataset**. Our method is trained based on HMR2.0 (Goel et al., 2023). PJ is short for PA-MPJPE. † indicates the upper bound of using GT 2D poses. The best and second best scores are highlighted, respectively. The numbers in brackets are improvements w.r.t. SOTA. Note that WHAM (Shin et al., 2024) also uses 2D poses, and ours outperforms it by $1.9mm$ PA-MPJPE.

| | METHOD | MPVPE↓ | MPJPE↓ | PJ↓ |
|---|---|---|---|---|
| IMAGE | SPIN (ICCV'19) | 112.8 | 96.9 | 59.2 |
| | I2L-MESHNET (ECCV'20) | - | 100.0 | 60.0 |
| | HYBRIK (CVPR'21) | 82.3 | 71.6 | 41.8 |
| | PYMAF (ICCV'21) | 110.1 | 92.8 | 58.9 |
| | PARE (ICCV'21) | 88.6 | 74.5 | 46.5 |
| | CLIFF (ECCV'22) | 81.2 | 69.0 | 43.0 |
| | BEDLAM-CLIFF (CVPR'23) | 85.0 | 72.0 | 46.6 |
| | REFIT (ICCV'23) | 75.1 | 65.3 | 40.5 |
| | TOKENHMR (CVPR'24) | 84.6 | 71.0 | 44.3 |
| | HMR2.0 (ICCV'23) | 82.2 | 69.8 | 44.4 |
| | **EXTPOSE** ($T = 1$) | **68.9** (-8.3%) | **55.6** (-14.9%) | **35.5** (-12.3%) |
| | **EXTPOSE**† ($T = 1$) | - | 36.7 | 25.4 |
| VIDEO | VIBE (CVPR'20) | 98.4 | 82.9 | 51.9 |
| | TCMR (CVPR'21) | 101.4 | 86.5 | 52.7 |
| | MAED (ICCV'21) | 92.6 | 79.1 | 45.7 |
| | MPS-NET (CVPR'22) | 99.0 | 84.3 | 52.1 |
| | GLAMR (CVPR'22) | - | - | 51.1 |
| | D&D (ECCV'22) | - | 73.7 | 42.7 |
| | SLAHMR (CVPR'23) | - | - | 55.9 |
| | TRACE (CVPR'23) | 95.4 | 79.1 | 50.9 |
| | GLOT (ICCV'23) | 96.3 | 80.7 | 50.6 |
| | WHAM (CVPR'24) | 68.7 | 57.8 | 35.9 |
| | **EXTPOSE** ($T = 16$) | **67.5** (-1.7%) | **54.2** (-6.2%) | **34.0** (-5.3%) |

based settings ($T = 1$). Given complicated light conditioning, occlusion, and truncation in 3DPW (Von Marcard et al., 2018), state-of-the-art (SOTA) HPE methods (Goel et al., 2023; Dwivedi et al., 2024b) struggle to predict accurate meshes based on the image's global context. In contrast, 2D pose estimators still provide more valuable localized cues, though SOTA is trained from ViTPose. Combining both modalities improves PA-MPJPE from $44.4mm$ to $35.5mm$, as shown in Tab. 1. With GT 2D poses, EXTPOSE matches the annotation errors of the dataset.

**Video benchmark** results for video-based HPE are also presented in Tab. 1. Existing methods typically rely on static image features, whereas our approach directly trains on image-based ViT methods, without additional modules. Contrary to prior trends where video-based methods (Shen et al., 2023) underperform image-based methods in framewise accuracy, our model achieves a $1.5mm$ improvement in PA-MPJPE by utilizing available video temporal context. EXTPOSE also outperforms WHAM by 5.3%, reflecting that our extended cross-modality and -frame attention derives a unified and superior framework.

### 5.3. Hand Pose Estimation

As our framework is widely applicable for both hand and human tasks, experiments are also conducted on the standard

Table 2: **SOTA 3D accuracy results on the FreiHAND dataset**. HaMeR (Pavlakos et al., 2024) is the ViT baseline. PJ and PV are short for PA-MPJPE and PA-MPVPE. The best and second best scores are highlighted respectively. The numbers in brackets are improvements w.r.t. SOTA.

| METHOD | PJ↓ | PV↓ | F@5↑ | F@15↑ |
|---|---|---|---|---|
| POSE2MESH (ECCV'20) | 7.7 | 7.8 | 0.674 | 0.969 |
| I2L-MESHNET (ECCV'20) | 7.4 | 7.6 | 0.681 | 0.973 |
| METRO (CVPR'21) | 6.5 | 6.3 | 0.731 | 0.984 |
| I2UV-HANDNET (ICCV'21) | 6.7 | 6.9 | 0.707 | 0.977 |
| HANDAR (ICCV'21) | 6.7 | 6.7 | 0.724 | 0.981 |
| MESHGRAPHORMER (ICCV'21) | 5.9 | 6.0 | 0.764 | 0.986 |
| MOBRECON (CVPR'22) | 5.7 | 5.8 | 0.784 | 0.986 |
| AMVUR (CVPR'23) | 6.2 | 6.1 | 0.767 | 0.987 |
| HAMER (CVPR'24) | 6.0 | 5.7 | 0.785 | 0.990 |
| **EXTPOSE** | **4.9** | **5.1** | **0.823** | **0.993** |
| | (-14.0%) | (-10.5%) | (+4.8%) | (+0.3%) |

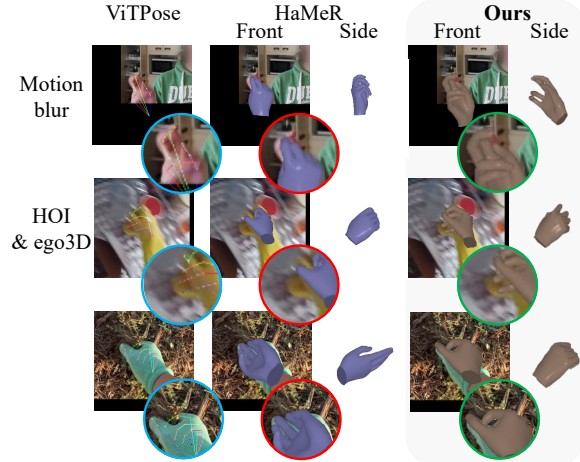

Figure 3: **Qualitative comparison.** The reconstructions are consistently enhanced, particularly on more challenging scenarios, *e.g.*, motion blur and (object and self-) occlusion.

hand benchmark to validate the method as follows.

**Settings.** For evaluation on hand video and image benchmarks below, we follow and also train HaMeR (Pavlakos et al., 2024) on multiple 3D datasets, FreiHAND (Zimmermann et al., 2019), HO3D (Hampali et al., 2020), MTC (Xiang et al., 2019), RHD (Zimmermann & Brox, 2017), InterHand2.6M (Moon et al., 2020), H2O3D (Hampali et al., 2020), DexYCB (Chao et al., 2021), and 2D datasets, COCO WholeBody (Jin et al., 2020), Halpe (Fang et al., 2022) and MPII NZSL (Simon et al., 2017).

**Image benchmark.** We evaluate our method with $T = 1$ on standard hand image datasets. **FreiHAND** (Zimmermann et al., 2019) includes diverse postures and severe self-occlusion. Despite the significant misalignment issues typical in generalization, few studies have explored 2D evidence for improving hand pose estimation (Figs. 1 and 3). Our results on EXTPOSE in Tab. 2 demonstrate the benefit of utilizing 2D grounding to enhance fair 3D accuracy benchmarks.

Table 3: **SOTA 3D metric results on the HO3D(v2) dataset**. HaMeR (Pavlakos et al., 2024) is the ViT baseline. PJ and PV are short for PA-MPJPE and PA-MPVPE. The best and second best scores are highlighted respectively. The numbers in brackets are improvements w.r.t. SOTA.

| | METHOD | AUC$_J$↑ | PJ↓ | AUC$_V$↑ | PV↓ | F@5↑ | F@15↑ |
|---|---|---|---|---|---|---|---|
| IMAGE | POSE2MESH (ECCV'20) | 0.754 | 12.5 | 0.749 | 12.7 | 0.441 | 0.909 |
| | I2L-MESHNET (ECCV'20) | 0.775 | 11.2 | 0.722 | 13.9 | 0.409 | 0.932 |
| | METRO (CVPR'21) | 0.792 | 10.4 | 0.779 | 11.1 | 0.484 | 0.946 |
| | LIU *et al.* (CVPR'21) | 0.803 | 9.9 | 0.810 | 9.5 | 0.528 | 0.956 |
| | I2UV-HANDNET (ICCV'21) | 0.804 | 9.9 | 0.799 | 10.1 | 0.500 | 0.943 |
| | ARTIBOOST (CVPR'22) | 0.773 | 11.4 | 0.782 | 10.9 | 0.488 | 0.944 |
| | KEYPOINTTRANS (CVPR'22) | 0.786 | 10.8 | - | - | - | - |
| | MOBRECON (CVPR'22) | - | 9.2 | - | 9.4 | 0.538 | 0.957 |
| | HANDOCCNET (CVPR'22) | 0.819 | 9.1 | 0.819 | 8.8 | 0.564 | 0.963 |
| | AMVUR (CVPR'23) | 0.835 | 8.3 | 0.836 | 8.2 | 0.608 | 0.965 |
| | HAMER (CVPR'24) | 0.846 | 7.7 | 0.841 | 7.9 | 0.635 | 0.980 |
| | **EXTPOSE** ($T = 1$) | **0.858** | **7.0** | **0.850** | **7.5** | **0.660** | **0.985** |
| | | (+1.4%) | (-9.1%) | (+1.1%) | (-5.1%) | (+3.9%) | (+0.5%) |
| VIDEO | VIBE* (CVPR'20) | - | 9.9 | - | 9.5 | 0.526 | 0.955 |
| | TCMR* (CVPR'21) | - | 11.4 | - | 10.9 | 0.463 | 0.933 |
| | TEMPCLR* (3DV'22) | - | 10.6 | - | 10.6 | 0.481 | 0.937 |
| | DEFORMER (ICCV'23) | - | 9.4 | - | 9.1 | 0.546 | 0.963 |
| | **EXTPOSE** ($T = 16$) | **0.863** | **6.9** | **0.856** | **7.3** | **0.667** | **0.991** |
| | | (-26.6%) | | (-19.8%) | (+22.2%) | (+2.9%) | |

Table 4: **SOTA 2D PCK results at different thresholds on the HInt dataset**. HaMeR is the ViT baseline. † indicates using the upper bound of GT 2D poses. The best and second best scores are highlighted respectively. The numbers in brackets are improvements w.r.t. SOTA.

| METHOD | NEW DAYS | | | VISOR | | |
|---|---|---|---|---|---|---|
| | @0.05↑ | @0.1↑ | @0.15↑ | @0.05↑ | @0.1↑ | @0.15↑ |
| METRO (CVPR'21) | 14.7 | 38.8 | 57.3 | 16.8 | 45.4 | 65.7 |
| FRANKMOCAP (ICCVW'21) | 16.1 | 41.4 | 60.2 | 16.8 | 45.6 | 66.2 |
| MESHGRAPHORMER (ICCV'21) | 16.8 | 42.0 | 59.7 | 19.1 | 48.5 | 67.4 |
| HANDOCCNET (CVPR'22) | 13.7 | 39.1 | 59.3 | 12.4 | 38.7 | 61.8 |
| HAMER (CVPR'24) | 48.0 | 78.0 | 88.8 | 43.0 | 76.9 | 89.3 |
| VITPOSE (NEURIPS'22) | 66.5 | 86.5 | 93.1 | 70.8 | 90.6 | 96.2 |
| **EXTPOSE** | **59.6** | **84.8** | **92.7** | **61.1** | **88.5** | **95.6** |
| | (+24.2%) | (+8.7%) | (+4.4%) | (+42.1%) | (+15.1%) | (+7.1%) |
| **EXTPOSE**† | 84.6 | 97.9 | 99.4 | 83.3 | 98.2 | 99.6 |

**Video benchmark.** HO3D is a hand-object interaction video dataset with occasional occlusions, where the space of poses is narrower than FreiHAND (Zimmermann et al., 2019). In addition to PA-MPJPE, HO3D benchmarks (Hampali et al., 2020) evaluate the Area Under the Curve for correct *3D* joints and vertices (**AUC$_J$**, **AUC$_V$**). As shown in Tab. 3, our method improves PA-MPJPE by 9.1% through better keypoint localization during object grasping, while further gains are achieved by leveraging information from other visual frames. Previous video-based HPE remains much less accurate than image-based methods, aligning with trends observed in human benchmarks.

**2D pose accuracy.** To assess real-world generalization, Pavlakos et al. (2024) introduced the challenging **HInt** dataset, capturing hands in complex daily activities from both exocentric and egocentric perspectives. While state-of-the-art HPE methods struggle, 2D pose estimation performs reasonably well (Tab. 4 and Fig. 3). This supports the idea that structured 2D pose localization can indeed improve

Table 5: **Ablations of branches and 2D pose representations on the FreiHAND dataset**. "Skel." is short for the skeleton.

| IMG | 2D POSE | PA-MPJPE↓ | PA-MPVPE↓ | F@5↑ | F@15↑ |
|---|---|---|---|---|---|
| | 1D | 6.5 | 6.6 | 0.724 | 0.983 |
| | HEATMAP | 6.3 | 6.3 | 0.747 | 0.984 |
| | SKEL. IMAGE | 6.2 | 6.3 | 0.742 | 0.985 |
| ✓ | | 6.0 | 5.7 | 0.783 | 0.991 |
| ✓ | SKEL. IMAGE | **4.9** | **5.1** | **0.823** | **0.993** |

HPE generalization in challenging scenarios. The accuracy drop from 2D projection, compared to ViTPose, is likely due to the generalization gap (similar to the GT 2D case), with model-based HPE as the primary focus, a trend also observed in (Chen et al., 2025a).

## 5.4. Ablations & Analysis

The ablation study is conducted for each component design one by one, on the in-distribution generalization of the Frei-HAND dataset (Zimmermann et al., 2019) and more challenging generalization on the HInt dataset (Pavlakos et al., 2024) and the 3DPW dataset (Von Marcard et al., 2018). Results across several hand and human benchmarks demonstrate the effectiveness and generality of our EXTPOSE framework.

**Optional 2D pose & quality.** We first assess the impact of 2D poses by training a variant with only the image and setting the 2D pose input to zero. As the ViT backbone is well-trained, this variant shows minimal performance improvement ($4^{th}$ row in Tab. 5), highlighting the value of precise 2D pose localization. Next, using ground-truth 2D poses instead of those estimated by Xu et al. (2022) further improves performance to an impressive $3.7mm$ on the FreiHAND dataset.

We also evaluate a lifting counterpart with only 2D pose input ($3^{rd}$ row in Tab. 5). In this case, 2D-to-3D lifting fails to leverage RGB image texture in situations of depth ambiguity or missing keypoint detection, underperforming image-based HPE. This reinforces the need for combining both modalities in HPE.

**2D pose representations.** Our key innovation is adopting an image-based representation of 2D poses, aligning more closely with RGB image features, and leveraging knowledge from pre-trained HPE models. For comparison with conventional 1D arrays, we employ a transformer encoder to extract features from joint coordinates, which are then fused with image features using cross-attention. Our image representation not only converges faster (see convergence study in App. C), but also achieves 4.6% lower PA-MPJPE (first three rows in Tab. 5). Spatial heatmaps outperform 1D arrays but are less preferred than skeleton images.

Table 6: **Ablations of different fusion and training strategies on the HInt dataset**. ∗ denotes adding extra parameters. "Only Q, K" indicates only training Q and K projectors of attention while "$1^{st}$ half" means only the first half of ViT blocks are trained. EXTPOSE features full attention and shares and trains full parameters.

| METHOD | NEW DAYS | | | VISOR | | |
|---|---|---|---|---|---|---|
| | @0.05 | @0.1 | @0.15 | @0.05 | @0.1 | @0.15 |
| HAMER | 48.0 | 78.0 | 88.8 | 43.0 | 76.9 | 89.3 |
| FUSION | | | | | | |
| LATE FUSION | 50.5 | 82.4 | 92.5 | 52.5 | 87.1 | 95.6 |
| CHANNEL CONCAT∗ | 56.3 | 83.6 | 92.2 | 55.9 | 87.3 | 95.3 |
| CONTROLNET∗ | 55.6 | 83.5 | 92.3 | 57.7 | 87.5 | 95.5 |
| TRAINING | | | | | | |
| FROM VITPOSE | 49.9 | 82.2 | 92.2 | 46.4 | 85.3 | 95.2 |
| ONLY Q, K | 50.0 | 81.9 | 92.3 | 49.1 | 85.3 | 95.1 |
| $1^{st}$ HALF | 50.8 | 82.2 | 92.3 | 50.2 | 85.8 | 95.2 |
| EXTPOSE | **59.6** | **84.8** | **92.7** | **61.1** | **88.5** | **95.6** |

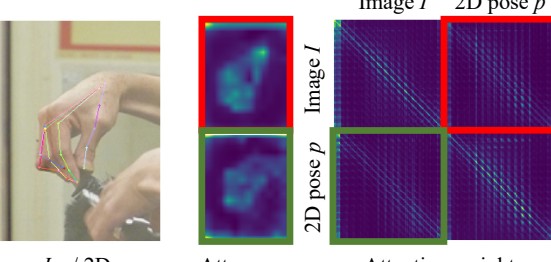

Figure 4: **Visualization of cross-modal attention** (Hertz et al., 2022). The attention weight is not solely concentrated along the diagonal, and the attention maps depict that each branch attends to distinct areas of the other branch, not just the center. Reference highlights on the corner and boundary indicate spatial awareness for 3D pose estimation.

**Image-2D pose fusions.** Apart from SOTA comparison with different fusion in Tab. 1, classic channel concatenation, late-layer fusion, and ControlNet fusion (Zhang et al., 2023a) are implemented under the same framework for a comprehensive comparison. Our attention mechanism enables more effective information exchange between the two streams at each layer, faster converging to improved performance ($2^{nd}$ block in Tab. 6). As shown in Fig. 4, cross-modal attention allows the branches to mutually enrich each other's representations. The image stream enhances alignment by querying clear *keypoint* location and structure information provided by the 2D hand skeleton image (**red** box) while the 2D pose stream *extensively* attends to missing RGB information across the hand and background (*e.g.*, wrist in the **green** box), aiding depth estimation.

**Inter-frame attention.** EXTPOSE with $T = 1$ improves pixel alignment with 2D pose assistance but causes jittering in video due to the lack of temporal frame relationships. Instead of retraining a video-based HPE model, video-based EXTPOSE leverages well-trained image-based models and extends attention across frames to reconstruct a smoother

Table 7: **Ablations of the sequence length on the 3DPW dataset (**Von Marcard et al., 2018**).**

| L | MVE↓ | MPJPE↓ | PA-MPJPE↓ |
|---|---|---|---|
| 1 | 68.9 | 55.6 | 35.5 |
| 8 | 68.2 | 55.1 | 34.7 |
| 16 | **67.5** | **54.2** | **34.0** |
| 32 | 67.9 | 54.8 | 34.3 |

4D mesh at minimal cost. For a better assessment, we provide rendered videos in the Sup. Mat. The sequence length hyperparameter is studied in Tab. 7, with 16 yielding reasonable results for the 3DPW dataset.

**Training strategies.** A key advantage of EXTPOSE is that it eliminates the need to train a model from scratch on large datasets. Table 6 ablates network initialization and learnable layers (Sec. 4.6), showing that initialization from ViT-based HPE methods and training the whole backbone is crucial for enhancing the model.

## 6. Conclusion & Limitations

ViT has been witnessed to be prevalent and dominant in HPE; yet, it is still not satisfactory with 2D image alignment and temporal coherence in real-world applications. In this work, a flexible framework EXTPOSE is proposed to extend the attention of SOTA image-based ViT HPE to encompass information of multiple additional dimensions. It benefits from elaborate representation unification, parameter reusing and sharing, and full hierarchical attention fusion. Extensive experiments demonstrate its wide effectiveness in rectifying visual alignment and capturing temporal relationships. In the future, reconstruction could be extended to cover the whole body. Additionally, it is also valuable to see the framework to take advantage of more visual modalities, such as SAM 2 prior (Ravi et al., 2025) and multiple views (Gao et al., 2024), as well as incorporating the latest Transformer efficiency studies.

## Impact Statement

This paper presents work whose goal is to advance the field of Machine Learning. There are many potential societal consequences of our work, none of which we feel must be specifically highlighted here.

## Acknowledgments

This research was also funded in part by a Google South Asia & Southeast Asia Research Award given to A. Yao and the National Natural Science Foundation of China (No.62406298). We would also like to thank the ACs and reviewers for their valuable suggestions. Chen and Rongrong contributed professional guidance in art design.

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

In addition to the self-contained manuscript, the appendix contains framework concept (App. A), more background (App. B), and experimental results (App. C).

## A. Extension Framework of Image-based Pose Estimation

As existing ViT-based methods typically require costly training, to make model upgrade more efficient and easier, a new paradigm EXTPOSE is proposed: extension of pre-trained image-based ViT HPE. Specifically, no additional parameter or retraining must be performed, and new 2D and temporal information are processed by a backbone shared with the image. It leverages flexible attention computation of ViT, which is not constrained to be identical to that in the pre-training. Therefore, the model easily extends to refer to newly provided information and adaptively captures mutual relationships. The new paradigm of our problem formulation also bridges relevant lifting and video-based HPE with image-based HPE and unifies them within a single cohesive framework. Notably, the extension paradigm is naturally driven by improvements in image-based HPE. This idea of clarity and practicality is inspired by other fields such as generation extending from the image (Esser et al., 2024) to current video (OpenAI, 2024) and 3D domain (Gao et al., 2024).

## B. 2D-to-3D Lifting

As mentioned in the related work, some works focus on the second stage of HPE, relying on 2D inference and totally ignoring the image. Among them, simple 2D joint coordinates are the most prevalent representation to work with, from which the 3D skeleton is lifted holistically. Martinez et al. (2017) establish a preliminary benchmark with a residual MLP in indoor human environments (Ionescu et al., 2013; Mehta et al., 2017). Followup works (Shan et al., 2022; Zhang et al., 2022; Zhu et al., 2023; Shan et al., 2023; Li et al., 2024) extend to explore the video setting and Transformer solution. Instead of outputting 3D joint coordinates, Choi et al. (2020) directly regresses the vertex coordinates of the mesh, eliminating an additional Inverse Kinematics task (Li et al., 2021).

**Comparison with our paradigm.** Conventional lifting and our work share similarities in supplementing models with 2D poses; both are evaluated with JPE-type errors. However, some critical protocol differences make the two lines of work difficult to compare directly. *Firstly*, *pose representation*, the target outputs in lifting are only skeleton joint coordinates, which favors the MPJPE metric (Yu et al., 2023). *Secondly*, lifting-based methods significantly benefit from using a larger *window size* (number of frames), typically 243, due to their operation in a lower-dimensional

Table 8: **SOTA 3D accuracy results on the Human3.6M dataset** ([Ionescu et al., 2013](#)). Our method improves HMR2.0 ([Goel et al., 2023](#)) with 16 frames and achieves comparable results to those using 243 frames. † denotes using GT 3D keypoints for final aggregation of multiple predictions. The best and second best scores are highlighted for lifting and model-based methods respectively. Note that methods are usually compared within their respective paradigms on this benchmark.

| | METHOD | $T$ | MPJPE↓ | PA-MPJPE↓ |
|---|---|---|---|---|
| LIFTING | TCN (CVPR'19) | 243 | 46.8 | 36.5 |
| | POSEFORMER (ICCV'21) | 81 | 44.3 | - |
| | ANATOMY (CSVT'21) | 243 | 44.1 | 35.0 |
| | STRIDED (TMM'22) | 351 | 43.7 | - |
| | MHFORMER (CVPR'22) | 351 | 43.0 | 34.4 |
| | MIXSTE (CVPR'22) | 243 | 39.8 | 30.6 |
| | P-STMO (ECCV'22) | 243 | 42.8 | 34.4 |
| | POSEFORMERV2 (CVPR'23) | 243 | 45.2 | 35.6 |
| | DIFFPOSE (CVPR'23) | 243 | 36.9 | 28.7 |
| | MOTIONBERT (ICCV'23) | 243 | 37.5 | - |
| | GLA-GCN (ICCV'23) | 243 | 44.4 | 34.8 |
| | D3DP† (ICCV'23) | 243 | 35.4 | 28.7 |
| | HOT (CVPR'24) | 243 | 39.0 | - |
| | FINEPOSE (CVPR'24) | 243 | 40.2 | 32.8 |
| | FINEPOSE† (CVPR'24) | 243 | **31.9** | **25.0** |
| MODEL-BASED | VIBE (CVPR'20) | 16 | 65.6 | 41.4 |
| | MEVA (ACCV'20) | 90 | 76.0 | 53.2 |
| | TCMR (CVPR'21) | 16 | 73.6 | 52.0 |
| | MPS-NET (CVPR'22) | 16 | 69.4 | 47.4 |
| | D&D (ECCV'22) | 16 | 52.5 | 35.5 |
| | GLOT (ICCV'23) | 16 | 67.0 | 46.3 |
| | HMR2.0 (ICCV'23) | 1 | 44.8 | 33.6 |
| | **EXTPOSE** | 16 | **43.5** (-2.9%) | **27.2** (-19.0%) |

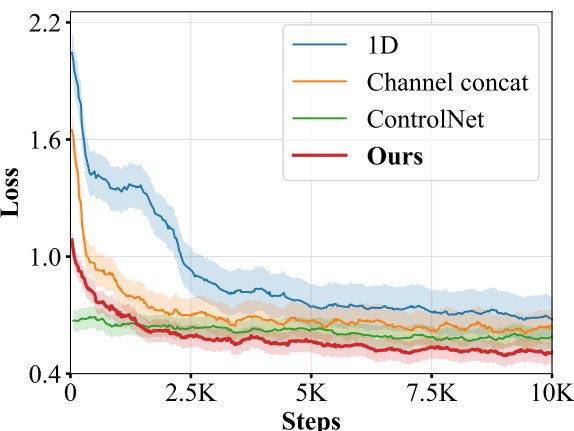

Figure 5: **Faster and more stable convergence compared to different 2D pose representations and fusion of hand experiments (Sec. [5.3](#)).** Solid lines and filled regions represent the mean and standard deviation respectively. ControlNet plug-in ([Zhang et al., 2023b](#)) is zero-initialized, resulting in a lower starting point.

space. Despite its simplicity, we qualitatively tested it and found that the lifting-based method falls short in the mesh reconstruction application and robustness on in-the-wild data. For respective quantitative results, please see App. [C](#).

## C. More Experimental Results

We present additional experiments with detailed captions, including quantitative results: **SOTA comparison on the Human3.6M dataset** ([Ionescu et al., 2013](#)) (App. [C](#)), **convergence plot for hand experiments** (Fig. [5](#)), **inference latency with extended attention** (Fig. [6](#)), and qualitative results: **with imperfect 2D poses** (Fig. [7](#)) and **challenging failures for future work** (Fig. [8](#)).

**Accuracy-efficiency tradeoff.** It is worth mentioning in Fig. [6](#) that the introduction of the 2D pose branch will roughly double the amount of data that needs to be processed; as keeping the total number of frames (batch size × sequence length) unchanged, the latency load brought by is not as obvious as it seems. Despite increasing computation, the overall efficiency is still acceptable to real-time.

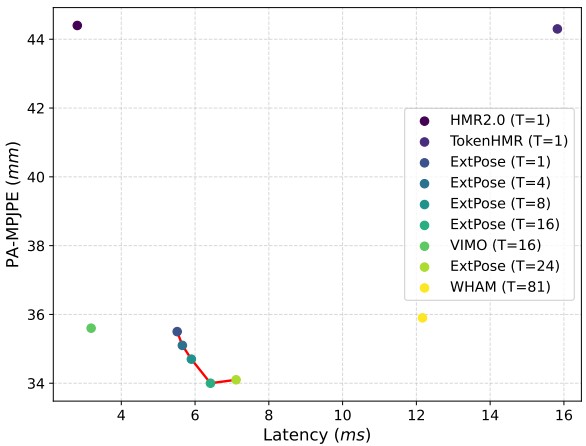

Figure 6: **The Efficiency-Accuracy tradeoff curve on the 3DPW dataset, *i.e.* Latency ($ms$)-PA-MPJPE ($mm$).** The per-frame computation time (running time) of core modules is measured with batch frames of 1024 on one A100 GPU. For the VIMO ([Wang et al., 2024](#)) result, time-consuming camera estimation is excluded. Please kindly note that the horizontal "Latency" axis itself does not scale proportionally.

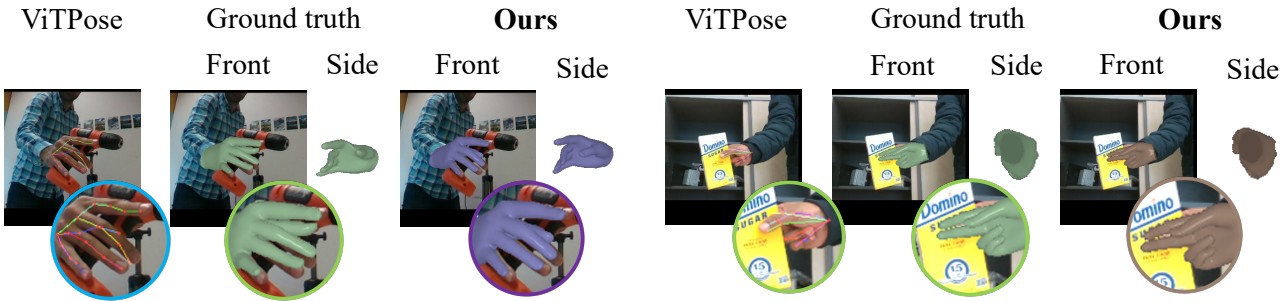

Figure 7: **Extended cross-modal attention can correct errors in ViTPose input.**

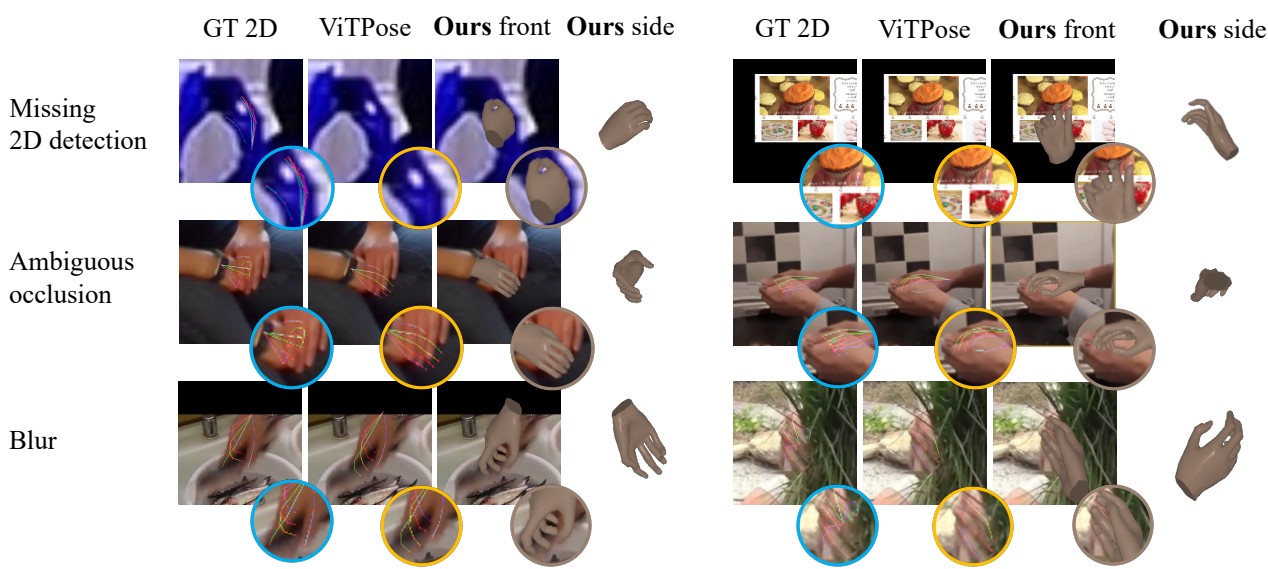

Figure 8: **Failures on include missing 2D keypoint detection, ambiguous occlusion, and blurred image.** These issues may be mitigated by leveraging more 2D prior knowledge, *e.g.* SAM 2 segmentation (Ravi et al., 2025).

