# OpenReview forum: "ExtPose: Robust and Coherent Pose Estimation by Extending ViTs"
_ICML.cc/2025/Conference — ICML 2025 poster_

### Official Review · Reviewer_AWPd · 2025-02-18

**Overall Recommendation:** 3

**Summary:**

This paper proposes a ViT-based model, ExtPose, for 3D single-human pose estimation. It can handle both image inputs and video inputs. Besides, it can utilize strong 2D HPE models (e.g. ViTPose) to enhance the 3D meshes. The overall performance is great - it achieves remarkable error reduction on existing benchmarks. ##update after rebuttal

**Claims And Evidence:**

The claims are well-supported.

**Essential References Not Discussed:**

I do not see a problem.

**Experimental Designs Or Analyses:**

The experiments are well-designed and persuasive. ExtPose achieves a remarkable error reduction compared to existing state-of-the-art methods. However, I think the reviewers should show the accuracy-efficiency trade-off curve in the paper - from existing tables, I cannot tell if the performance mainly comes from more computation. Also, I noticed ExtPose still cannot outperform lifting-based methods in Table 8. I wonder the reason for that. Another minor novelty issue could be that cross-frame attention is a common technique in other areas [1].

[1] Gao, Ruiqi, et al. "Cat3d: Create anything in 3d with multi-view diffusion models." arXiv preprint arXiv:2405.10314 (2024).

**Methods And Evaluation Criteria:**

The benchmarks and metrics used are standard.

**Other Comments Or Suggestions:**

N/A

**Other Strengths And Weaknesses:**

N/A

**Questions For Authors:**

N/A

**Relation To Broader Scientific Literature:**

Human Pose Estimation/AR/VR.

**Theoretical Claims:**

I do not see a problem.

---

> ### Author Rebuttal · Authors · 2025-03-31
>
> We express our sincere appreciation for the helpful reviews and tackle the concerns below:
>
> **[Design 1] Accuracy-efficiency tradeoff curve?**
>
> **A:** Thanks for this suggestion. Table 9 in our submission already follows Xu et al. (2022) and Wang et al. (2024) to demonstrate computational efficiency. For clarity, we plot them along with accuracy in Fig. a ([anonymous link](https://anonymous.4open.science/r/ExtPose_ICML25/icml2025_extpose_2283.pdf)).
> As the amount of attention computation increases with the addition of 2D pose modality and the number of frames, the error does decrease steadily, but the accuracy gain brought by the increase in computation gradually diminishes until the $T=16$ plateau, so it is necessary to weigh the actual accuracy against the efficiency required. It is worth mentioning that the introduction of the 2D pose branch will roughly double the amount of data that needs to be processed; as keeping the total number of frames (batch size$\times$sequence length) unchanged, the latency load brought by $T$ is not as obvious as it seems. Despite of increasing computation, the overall efficiency is still acceptable to real-time. In this work, our primary goal is to achieve high accuracy, and we also mentioned that efficiency could be improved in future work.
>
> **References:**
> - Wang et al. YOLOv10: Real-time end-to-end object detection. NeurIPS 2024.
> - (VIMO) Wang et al. TRAM: Global trajectory and motion of 3D humans from in-the-wild videos. ECCV 2024.
>
> **[Design 2] ExtPose underperforms lifting-based methods?**
>
> **A:** Thanks for your careful review. A short answer is, model- and SOTA lifting-based paradigms are usually not fairly compared due to **1.** different representations, **2.** the use of 3D poses, and **3.** window sizes. We outperform them in a similar setting (R1 uud2-Tab. a) and our ablation study (Tab. 5). We kindly refer the reviewer to check our response to **R1 uud2-Question 2** in detail. We will also make the point clearer in the revision.
>
> **[Design. 3] Cross-frame novelty.**
>
> **A:** Thanks for your recommendation and comments. Our contribution lies in inspecting and effectively solving the challenges of 2D misalignment and temporal inconsistency of current ViT human pose estimation models with a simple and elegant solution. We devise a unified 2D pose representation with attention extended across the modality and frame to derive a robust video ViT model.  The attention is thus implemented “simply and straightforwardly”, which aligns with techniques applied in other fields such as Multiview 3D generation, as the reviewer kindly mentioned. We will also include it in the reference.
>
> We believe your contribution will definitely help enhance our work. Please feel free to let us know if there are still concerns not addressed in our feedback, and we are more than happy to communicate.

---

> > ### Comment · Reviewer_AWPd · 2025-04-06
> >
> > Thank you for the rebuttal! Design 1 addresses my concern, but the other two do not persuade me. I will keep my initial score.

---

> > > ### Author Response · Authors · 2025-04-07
> > >
> > > We are pleased to address the reviewer's concern and appreciate the discussion on the lifting-based method and cross-frame attention novelty.
> > >
> > > Our method outperforms all lifting-based methods including FinePOSE in similar settings, and we also highlight the differences between these two paradigms. Despite its simplicity, the lifting-based method falls short in the mesh reconstruction application and robustness on in-the-wild data. We tested their official demos on the Sup. video and found that the overall performance was not satisfactory.
> > >
> > > Additionally, we would like to highlight that our attention is flexible and extendable for pose estimation tasks. It is promising to incorporate multi-view and more various modalities in the future.
> > >
> > > Again, we are grateful for the reviewer's thorough review and dedication to our work.

---

### Official Review · Reviewer_JotU · 2025-03-06

**Overall Recommendation:** 4

**Summary:**

The authors propose ExtPose: a robust and Coherent pose estimation by refining a ViT-based HPE. Several contributions are proposed in this paper: 1) 2D pose and image information are combined are combined in a ViT model, 2) Temporal context is integrated. The resulting model is compared to the SOTA models using 3DPW dataset. The proposed model outperforms the SOTA models with a large margin for some metrics. The ablation study shows that each component of the proposed model is effective. The paper is well written and the experiments are well conducted. This paper is a good contribution to my point of view.

**Claims And Evidence:**

- The authors propose to use a skeleton-based image representation in a ViT model (inspired from Zhang et al.). This is a good idea to combine 2D pose and image information in a ViT model.
- Taking into account the temporal context is done by spatiotemporal attention on features between frames.
- Experiments have been conducted on several datasets and the proposed model outperforms the SOTA models.
- The experiments show that the proposed model outperforms the SOTA models with a large margin for some metrics.
- The ablation study shows that each proposed contribution is effective and improves the performance of the model.

**Essential References Not Discussed:**

/

**Experimental Designs Or Analyses:**

/

**Methods And Evaluation Criteria:**

/

**Other Comments Or Suggestions:**

/

**Other Strengths And Weaknesses:**

/

**Questions For Authors:**

/

**Relation To Broader Scientific Literature:**

/

**Theoretical Claims:**

/

---

> ### Author Rebuttal · Authors · 2025-03-31
>
> We are grateful for the reviewers' valuable efforts and their recognition of our work! If you have any other questions, we are always here, and you are welcome to discuss with us.

---

### Official Review · Reviewer_Sqcd · 2025-03-12

**Overall Recommendation:** 3

**Summary:**

The paper proposes a 3D pose estimation algorithm that simultaneously considers 2D pose information and temporal information. Through parameter sharing and multimodal data alignment strategies, the algorithm is able to accurately estimate 3D pose. The authors validate their proposed approach through extensive experiments, achieving state-of-the-art (SOTA) results across nearly all evaluation datasets and metrics.

## update after rebuttal
Thanks for the clarification. After reviewing all the feedback and rebuttals, I’ll be keeping my score as is. However, I hope the authors can address the concerns raised by others and include the necessary details in the revised version.

**Claims And Evidence:**

The authors identify three key design factors that significantly impact the final results: consistency in the representation space, the use of parameter sharing, and the hierarchical attention fusion strategy. These factors are validated in the subsequent experiments.

**Essential References Not Discussed:**

Yes, this paper has discussed enough related references.

**Experimental Designs Or Analyses:**

The experimental design in the paper is quite reasonable, and the algorithm's effectiveness is validated from different dimensions. Since I am not an expert in this field, I have one question: the method requires additional 2D pose as input. If the 2D pose detected by the algorithm contains significant noise, can it be corrected in the subsequent stages of the algorithm?

**Methods And Evaluation Criteria:**

The metrics used in the paper are standard, and when comparing with related work, reasonable common metrics are employed.

**Other Comments Or Suggestions:**

A small issue is that the paper repeatedly emphasizes "extending" ViT or poses. Could the title also use "extending" instead of "expanding" to maintain consistency?

**Other Strengths And Weaknesses:**

The contributions presented in the paper are validated in subsequent experiments, and compared to previous work, the results achieve the best performance across nearly all metrics. I believe this is a meaningful contribution. However, I have two concerns: first, the reliance on the output 2D pose, could you clarify how much this dependence impacts the results? Second, the discussion on failure cases and limitations, some frames in the supplementary materials do not produce a reasonable mesh, with a significant deviation from the image. Could you provide an explanation for this?

**Questions For Authors:**

As discussed above, I hope the authors can address the issues I have mentioned above.

**Relation To Broader Scientific Literature:**

The paper observes the limitations of previous work in addressing temporal images and proposes corresponding strategies to resolve these issues. It also introduces 2D pose as a prior, further improving accuracy. This is a contribution to the research direction in this area.

**Theoretical Claims:**

The theoretical explanations in the paper are reasonable, and the presented results align with the theoretical background.

---

> ### Author Rebuttal · Authors · 2025-03-31
>
> We sincerely appreciate the reviewer's constructive comments and address the concerns below:
>
> **[Strengths & Weaknesses 1/Exp. 1] The effect of noise in 2D pose detection?**
>
> **A:** Thanks for your great insight in studying the effect of 2D pose quality on performance!
> - Firstly, in Fig. 6, we **qualitatively** highlight when erroneous 2D poses occur (_e.g._ left middle and right ring fingers), the original image branch assists in maintaining accurate predictions, _i.e._ correcting the 2D pose inaccuracy as the review rightly inferred. But indeed, there are failures where the prediction still adheres to the wrong 2D pose guess, particularly when the image information is hard to perceive in challenging cases, _e.g._ occlusion and blur in Fig. 7.
> - Secondly, to **quantitatively** assess the effect of 2D pose noise, we introduce Gaussian noise with increasing standard deviations to the 2D pose while following Gu et al. (2024) to adjust the confidence score accordingly to reflect the increasing prediction uncertainty. It is worth mentioning that not only the coordinate prediction but also confidence as additional information are drawn in the 2D pose image. Table b shows the benefit from auxiliary 2D poses diminishes progressively with the increase in noise levels. When the confidence becomes low and the keypoint is displayed as transparent in the 2D pose image, the model does not use much of 2D poses and degenerates to the original ViT model (instead of collapsing), indicating the robustness to 2D pose noise of varying levels.
>
> **Table b. The effect of synthetic 2D pose noise on the 3DPW image dataset.** ExtPose makes use of high-quality 2D pose detection while maintaining robustness to pose noise.
> | Noise (pix) | PA-MPJPE | MPJPE |
> | --- | :---: | :---: |
> | std = 0	| 35.5 | 55.6 |
> | std = 4	| 38.5 | 62.3 |
> | std = 8	| 43.3 | 69.0 |
> | std = 12	| 44.2 | 69.5 |
> | HMR2.0 	| 44.4 | 69.8 |
>
> **References:**
> - Gu et al. On the calibration of human pose estimation. ICML 2024.
>
> **[S & W 2/Sup. 1] Explanation for the failure in the Sup. video?**
>
> **A:** Thanks for the insightful observations. Yes, although our method shows significant enhancement in robustly aligned and temporally coherent pose estimation, challenges remain. For instance, as mentioned earlier, Fig. 7 showcases some really challenging and long-tail scenarios like ambiguous occlusion and blur. Analogously, the hand misalignment in the Sup. video frame, we speculate, likely stems from 2D pose estimation difficulties and, therefore, are not confident to provide any useful cues (not depicted in the pose image). Such out-of-distribution cases can lead to diverse error predictions across different models. A thought of future work to mitigate the issue may be to leverage more 2D prior knowledge, _e.g._ SAM2 segmentation.
>
> **Other writing suggestions** will be fixed accordingly. Thanks!
>
> We are confident that your input will greatly improve our manuscript. Should there be any unresolved issues in our feedback, please do not hesitate to inform us, as we are committed to continuous improvement and welcome further dialogue.

---

### Official Review · Reviewer_uud2 · 2025-03-13

**Overall Recommendation:** 3

**Summary:**

This work presents ExtPose, a ViT-based framework that extends a pre-trained ViT backbone to better handle image alignment and temporal coherence by introducing the following: 2D skeleton images as additional input, cross-modality interaction, and cross-frame interaction. Even though the proposed methods can be implemented efficiently, validation in multiple downstream tasks shows that ExtPose can outperform the baseline.


## Update after Rebuttal

After the rebuttal, the contribution of the paper seems solid and the performance of the proposed method is strong. Therefore, I have raised my score to 3, and I lean towards accepting this paper. However, I strongly recommend that the authors incorporate the feedbacks from the reviewers.

**Claims And Evidence:**

The effectiveness of the presented approach has been validated by the evaluation of multiple downstream tasks, improving the performance over the baseline.

**Essential References Not Discussed:**

I did not find any crucial references not discussed in the manuscript.

**Experimental Designs Or Analyses:**

The visualization and analysis of the cross-modal attention do not seem to be very convincing. Specifically, the following are some of the questions I would like the authors to clarify:

1. Why is the attention map of the first element (the top left position) always show dominant weights in all four quadrants? This seems to show some kind of positional bias.
2. The patterns in the attention weights do not seem very informative. I do not understand how this attention weight visualization can be connected to the authors' claim that "The image stream queries clear location and structure information provided by the 2D hand skeleton image (red box) while the 2D pose stream attends to missing RGB information across the hand and background", as it simply seems to be attending to most of the parts in the middle of the image, where the 2D pose skeleton will mostly appear.

**Methods And Evaluation Criteria:**

For most of the parts, the explanation of the method is easy to understand, however, I am quite confused of how the attention mask operates for the operation "Cross-Frame Attention" in Section 5.4.
Specifically, why is the pose-pose feature attention part added with a value of one whereas the image-image feature attention part is added with zero? I would like an extra explanation or justification for this choice.

**Other Comments Or Suggestions:**

1. There is a missed period on L203, "This underscores ViT’s generalizability and scalability while demonstrating that 2D pose features
 extracted this way align well with image features".

2. I would like to encourage the authors to simply describe Human Pose Estimation and Hand Pose Estimation in their full names. Although they have provided a subscript that HPE will be used interchangeably, I do not find a significant reason to do this, and it only increases confusion.

**Other Strengths And Weaknesses:**

Although the technical contribution is not novel or significant, I believe that introducing the concept of 3D attention together with the 2D pose map is interesting and effective for improving the overall performance.

**Questions For Authors:**

1. I would like the clarification of the visualization of the attention maps in Figure 4, as I have explained in "Experimental Designs Or Analyses*
".

2. I have a question regarding the evaluation shown in Table 8, which was my primary justification for giving a "Weak Reject" as my initial rating for this paper. It seems that the authors are considering Lifting-based methods not as their primary competitors but I did not understand why this is the case. As this framework also utilizes 2D poses, I am not fully sure why methods such as FinePose should not be discussed, even though they perform much better than the proposed method ExtPose. If the authors can justify why these methods should not be compared directly (i.e., additional ground truths, and different training data), I would like to raise my score.

**Relation To Broader Scientific Literature:**

This work directly builds upon a recent framework and as the proposed framework is easy to implement, I believe that a similar approach can be applied to future improvements.

**Theoretical Claims:**

There do not seem to be any theoretical claims in this work.

---

> ### Author Rebuttal · Authors · 2025-03-31
>
> We sincerely thank the reviewer for the valuable feedback and address the concerns below:
>
> **[Method 1] Pose-pose attention mask in Eq. (11)?**
>
> **A:** Really thanks for your careful inspection! There is indeed a typo in Eq. (11). As described in the text Ls. 223-229 right above Eq. (11), the lower right quadrant pose-pose attention should also be a full 0 matrix. This is to focus on temporal modeling only within the respective modal branch.
>
> **[Question 1.1/Exp. 1] In Fig. 4, the highlight in the left corner indicates positional bias?**
>
> **A:** This is really an insightful observation! The phenomenon is also observed in other ViT-based methods like HMR2.0. Indeed, we speculate this is expected as the image corner and boundary may serve as helpful references for global/absolute positioning and final SMPL parameter regression. Thus, there are highlights indicating spatial awareness and being crucial for accurate pose estimation.
>
> **[Question 1.2/Exp. 2] How to interpret mutual attention of two branches from Fig. 4?**
>
> **A:** Thank you for highlighting the unclear points!
>
> Column meanings: In the $3^{rd}$ column “Attention weights,” row $x$ inside the red box shows the flattened attention between point $x$ in the image and the entire 2D pose image. The red box in the $2^{nd}$ column “Attn maps” averages across the row (_i.e._, all points in the image branch) and reshapes to display the image branch's attention to the 2D pose for information gathering. The green boxes depict the reverse process: where the 2D pose focuses on the image.
>
> Thus, the claim is supported by **keypoint** emphasis in the upper figure and **extensive** attention across the lower figure. Specifically, the upper figure in the middle column highlights localized 2D keypoint locations, explaining improved alignment; the lower figure reveals that 2D pose tokens focus on **not only the central foreground hand but also the background** (_e.g._, wrist in the green box, in Ls. 399-401 right), collecting RGB info and aiding 3D lifting. This broader focus helps estimate depth using the additional contextual info.
>
> We will supply Fig. 4 with more details in our revision.
>
> **[Question 2] Comparison to lifting-based methods, specifically FinePOSE the only one better?**
>
> **A:** Thanks for raising this point! Indeed, lifting and our work share similarities of supplementing models with 2D poses; both are evaluated with JPE-type errors.  However, some critical protocol differences make the two lines of work difficult to compare directly.
> - Firstly, the target outputs in lifting are only skeleton joint coordinates.  In model-based human mesh reconstruction (HMR) which also estimates shape, the output is SMPL parameters or joint rotations.  The paradigm of direct coordinate regression has more advantage in body position prediction than SMPL-based methods due to the bottleneck of SMPL representation (Yu et al., 2023), thus tending to have a lower MPJPE (Tab. 8).
> The _MPJPE_ metric has different preferences of different **pose representations**.
> - Secondly, FinePOSE (and other SOTA lifting methods like D3DP) tend to use **GT 3D** keypoints for final aggregation of its multiple predictions; this is the case of their SOTA performance of 25.0mm in _PA-MPJPE_.  Without GT 3D keypoints, their performance drops to 32.8mm (vs. our 27.2mm shown in Tab. a $2^{nd}$ row). Please find more details of the J-Best and J-Agg metric in the D3DP paper.
> - Thirdly, lifting-based methods significantly benefit from using a larger **window size** (number of frames), specifically 243, due to their operation in a lower-dimensional space. This advantage is evident as shown in the 3$^{rd}$ row of Table a, where a reduction in window size to 16, typical of model-based methods constrained by available GPU memory, results in a noticeable increase in PA-MPJPE (Pavllo et al., 2019; Zhang et al., 2022).
>
> To make a more equal comparison, we add ablation studies in Tab. 5 and Ls. 382-384 right & 412-414 left: our ExtPose using both the image and 2D pose outperforms either the image or 2D pose branch alone (_i.e._ 2D-to-SMPL lifting) in our setting. We will add the context and description in the revision.
>
> **Table a. Comparisons with the SOTA lifting-based method on the Human3.6M dataset.** “Agg.” stands for aggregation. With **2)** no GT poses and **3)** a same window size of 16, ours outperforms FinePOSE.
>
> | Method | #Frames | MPJPE | PA-MPJPE |
> | --- | :---: | :---: | :---: |
> | FinePOSE* (GT Agg.) 	| 243 | 31.9 | 25.0 |
> | FinePOSE 			| 243 | 40.2 | 32.8 |
> | FinePOSE			| 16 | 50.4 | 41.0 |
> | **ExtPose** 			| 16 | 43.5 | 27.2 |
>
> **References:**
> - Yu et al. Overcoming the trade-off between accuracy and plausibility in 3D hand shape reconstruction. CVPR 2023.
>
> **Other writing suggestions** will be revised accordingly, and thanks for bringing these to our attention.
>
> Please feel free to point out any unclear issues, as we value your feedback highly and are always ready to discuss them further.

---

> > ### Comment · Reviewer_uud2 · 2025-04-05
> >
> > Thank you for addressing my questions. After reviewing the rebuttal from the authors, my major concerns or questions are addressed. In this point, I am leaning towards accepting this paper and will raise the score to 3.

---

> > > ### Author Response · Authors · 2025-04-05
> > >
> > > We are grateful that our feedback has successfully addressed the reviewer's concerns, leading to a recommendation to accept our work. _To summarize_, we appreciate the _recognition_ of our work’s interesting concepts, effectiveness contribution to the field, sufficient evaluation, and its clarity and practicality, etc. Additionally, we value the insightful _suggestions_ for improvements, including clearer explanations of our comparison setting with lifting-based methods and attention maps. Sincerely, we thank the reviewer for the invaluable expertise and guidance in enhancing our manuscript. We commit to diligently incorporating the feedback in the revised manuscript.

---

### Decision · Program_Chairs · 2025-05-01

**Decision:**

Accept (poster)

**Comment:**

This work introduces a 3D pose estimation method considering 2D pose information together with temporal information. The proposed method can estimate 3D pose reliably, by sharing parameters and aligning multi-modal data. The authors have done extensive experiments showing the efficacy of the method. Overall, the method is interesting and reasonable. All the reviewers recommend acceptance. AC agrees with reviewers.